# Silk Fibroin-Based Therapeutics for Impaired Wound Healing

**DOI:** 10.3390/pharmaceutics14030651

**Published:** 2022-03-16

**Authors:** Tanner Lehmann, Alyssa E. Vaughn, Sudipta Seal, Kenneth W. Liechty, Carlos Zgheib

**Affiliations:** 1Laboratory for Fetal and Regenerative Biology, Department of Surgery, University of Colorado Denver School of Medicine, Aurora, CO 80045, USA; tanner.lehmann@cuanschutz.edu (T.L.); alyssa.vaughn@cuanschutz.edu (A.E.V.); kenneth.liechty@childrenscolorado.org (K.W.L.); 2Division of Pediatric Surgery, Children’s Hospital Colorado, Aurora, CO 80045, USA; 3Department of Material Science and Engineering, Advanced Materials Processing and Analysis Center, Nanoscience Technology Center, University of Central Florida, Orlando, FL 32827, USA; sudipta.seal@ucf.edu; 4UCF Prosthetics Cluster, College of Medicine, University of Central Florida, Orlando, FL 32827, USA

**Keywords:** silk fibroin, wound healing, nanosilk, diabetes, nanotechnology, nanoparticles, cerium oxide

## Abstract

Impaired wound healing can lead to local hypoxia or tissue necrosis and ultimately result in amputation or even death. Various factors can influence the wound healing environment, including bacterial or fungal infections, different disease states, desiccation, edema, and even systemic viral infections such as COVID-19. Silk fibroin, the fibrous structural-protein component in silk, has emerged as a promising treatment for these impaired processes by promoting functional tissue regeneration. Silk fibroin’s dynamic properties allow for customizable nanoarchitectures, which can be tailored for effectively treating several wound healing impairments. Different forms of silk fibroin include nanoparticles, biosensors, tissue scaffolds, wound dressings, and novel drug-delivery systems. Silk fibroin can be combined with other biomaterials, such as chitosan or microRNA-bound cerium oxide nanoparticles (CNP), to have a synergistic effect on improving impaired wound healing. This review focuses on the different applications of silk-fibroin-based nanotechnology in improving the wound healing process; here we discuss silk fibroin as a tissue scaffold, topical solution, biosensor, and nanoparticle.

## 1. Introduction

Skin wounds are a natural part of life; therefore, organisms are generally well prepared to repair the resulting damaged tissue. When a wound occurs under normal, healthy circumstances, an extraordinarily complex and delicate process is initiated to reverse the damage [1]. The wound repair process consists of four overlapping phases: initial hemostasis, the inflammatory phase, the proliferation phase, and the maturation phase [2]. Each phase employs numerous cells, cytokines, and growth factors to facilitate the structural repair and closure necessary to restore tissue to its undamaged state. Tissue regeneration can quickly become deranged if one or more of these components is dysregulated. Dysregulation can occur for many reasons; chronic conditions such as diabetes mellitus, cancer, malnutrition, and even sequelae of COVID-19 can each lead to the development of chronic wounds [3]. In 2014 alone, it was estimated that Medicare patients spent up to 96.8 billion USD on wound care [4]. It is estimated that more than 29 million people in the United States alone—nearly 1-in-10—have been diagnosed with diabetes mellitus, with rates continuing to rise [5]. Worldwide, it is estimated that 552 million people will have diabetes by 2030 [6]. The increasing prevalence of individuals with impaired wound healing and the substantial associated costs illustrate the need for cost-effective treatment options.

The ideal wound healing treatment is composed of highly biocompatible, bioactive materials that aid in wound closure, degrade or metabolize at an appropriate rate, protect the wound from microbial infection, can be removed without damaging the underlying tissue, and are easily attainable. However, traditional wound healing treatments, such as skin autografts, can be costly, invasive, inefficient, and even harmful to the wound [7]. There are currently no widely-used wound treatments that reach these ideal standards. Biomaterials such as collagen, elastin, and gelatin have all been studied as wound treatments due to their presence in nature. Ultimately, many of these biomaterials are hindered by processing limitations, structural degradation, and by their immunogenicity in wounds.

Silk fibroin (SF) has emerged as a dynamic biomaterial that meets the aforementioned criteria for an ideal wound treatment. SF is derived from cocoons of *Bombyx mori* silk worms and also from the webs spun by spiders, mites, and other insects [8,9]. Silk contains two proteins: SF and sericin. SF is a fibrous structural protein and is the component typically isolated for therapeutic applications. Sericin creates a gum-like structure of glycoproteins surrounding the unprocessed SF and can exhibit immunogenicity; therefore, it is usually removed [9]. In order to separate these two elements, sericin is removed via a degumming process, and the remaining SF is regenerated via electrospinning, a fairly simple and inexpensive process. SF’s dynamic properties as a biomaterial can be applied to many wound treatment approaches, such as molecular scaffolds, topical applications, and novel therapeutic delivery systems [10]. This natural bioactive polymer is effective, readily available, and cost effective, making it an ideal candidate for widespread usage. While SF has been studied as a regenerative therapeutic for a plethora of tissues, including bone, cornea, nerve, and cartilage, this review focuses on the use of SF as it relates to cutaneous wound healing and tissue engineering.

## 2. Physicochemical Properties of Silk Fibroin

SF has been widely researched as a wound therapeutic due to its dynamic properties and biocompatibility. Both the physical and chemical properties of this natural biopolymer lend to the benefits of using SF as a wound treatment. However, there are some disadvantages to SF, which are discussed as well.

Three components make up SF: a heavy chain, a light chain, and a glycoprotein, which are 350 kDa, 25 kDa, and 30 kDa, respectively [11]. The heavy chains consist of hydrophobic domains, while the light chains are hydrophilic. These two chains form both secondary structures commonly seen in SF: silk I and silk II [12]. Silk I forms α-helices, while silk II forms β-sheets. In particular, the β-sheets form hydrogen bonds and, alongside glycine and alanine bonds, lend the biomaterial its renowned mechanical strength. Genetic manipulation of SF can lead to tunable properties which affect how the biomaterial behaves [13]. These tunable properties include permeability, composition, and sequence.

Ultimately, the physicochemical properties of SF combine to create an ideal biomaterial for wound treatment. High water retention keeps the wound hydrated, while antimicrobial properties prevent infection [7]. Improved cytocompatibility and efficient carbon dioxide and oxygen gas exchange allow cells to more efficiently proliferate within the wound. Little to no immunogenicity keeps inflammation low, reducing the risk of adverse reactions [7]. Additionally, it can work synergistically when used in conjunction with other biomaterials such as chitosan or microRNA-conjugated cerium oxide nanoparticles (CNP) [14].

However, SF also has its disadvantages (Table 1). The cross-linking of the β-sheets that confer mechanical strength can be vulnerable to enzymatic degradation. Matrix metalloproteinases (MMPs) have been shown to degrade SF in solution. Proteinase K has high affinity for the β-sheet component of SF, and collagenase degrades the amorphous regions [15]. Given that chronic wounds typically over-express various proteases, it is important to consider degradability when using SF as a therapeutic [16]. Some research suggests that tightly conformed β-sheets have significant resistance to enzymatic degradation [17]. Additionally, SF hydrogels have been shown to have poor mechanical strength and can undergo swelling in the wound [11]. Mechanical strength is a central feature of SF as a wound treatment, and the loss of this greatly reduces effectivity. Fortunately, there are solutions to this. Combining SF hydrogels with various polymers can greatly improve SF-hydrogel properties [11,18].

## 3. SF Scaffolds

The regeneration of the extracellular matrix (ECM) after tissue is wounded is a critical component of restoring skin to its healthy state [28]. The ECM is found in all tissues and organs and plays a role in regulating nearly all cellular functions and tissue morphologies by binding growth factors and mediating signal transduction [29]. Importantly, it also acts as a structural scaffold for cell adhesion and migration. The fibrous network primarily consists of water, glycosaminoglycans, proteoglycans, and fibrous proteins like collagen and elastin; the ECM is exceptionally dynamic, and its exact composition varies in a tissue-specific manner [30]. In the skin and other connective tissues, dermal fibroblasts are responsible for secreting fibrous proteins and integrating them into the ECM. These fibrous proteins provide the biophysical component of the ECM, forming rope-like structures and sheets that organize into non-cellular scaffolds [30].

Due to the importance of the ECM to the integrity of a tissue, many wound treatments under development are aimed at creating a biomimetic ECM or tissue scaffold. Specifically, tissue scaffolds are bioengineered skin substitutes used to mimic the nanoarchitecture and functionality of a tissue’s endogenous ECM [31]. Aptly named, tissue scaffolds provide a three-dimensional structure to which cells can adhere and subsequently repair wounds. The ideal tissue scaffold retains water, allows for adequate gas exchange, and improves cell adhesion and motility. It also has the ability to deliver drugs or other treatments, biodegrade, be non-immunogenic, and interact with the existing ECM. It is also important for these scaffolds to be easily manufactured and cost effective, given the increasing frequency of diseases leading to impaired wound healing.

Collagen is the most prolific protein in animals and plays essential roles in wound healing such as providing tensile strength and scaffolding to healing tissue and supporting chemotaxis and the migration of necessary cells. Naturally, as the primary structural component of the ECM, collagen has been extensively researched as a potential tissue scaffold to treat cutaneous wounds and is currently the most commonly used biomaterial for treating wound healing [7]. However, the use of ECM proteins such as collagen, fibrin, and elastin as therapeutic biomaterials has presented certain challenges [7]. Endogenous mechanisms can counteract the use of collagen in chronic wounds. During the inflammation phase of wound healing, matrix metalloproteases (MMPs)—a large family of ECM-degrading proteases—break down the damaged ECM in order for new matrices to be added [1,32]. As the most abundant protein, collagen is an obvious target of multiple collagenases (MMP-1, MMP-8, and MMP-13), which therefore limits the use of additional collagen as a tissue scaffold [33].

Additionally, despite extensive research, practical issues with manufacturing collagen remain unsolved [7]. For example, collagen loses its critical cross-linked structure during processing and isolation. The primary benefit of collagen scaffolds is their biophysical strength, especially in wound beds with insufficient tensile resistance [34]. If this benefit cannot be retained throughout the processing needed for its use as a biomaterial, it cannot be the gold standard for wound treatments. Further, collagen is typically derived from animals and has been known to carry pathogens, which can induce immunogenicity in sensitive patients and exacerbate poor wound healing conditions [35]. Additionally, collagen is cost-prohibitive and more difficult to obtain than other tissue-engineered biomaterials such as silk. Unless these issues can be resolved, collagen is not ideal for widespread use, and other biomaterials should be explored.

SF uniquely solves many of the issues presented with collagen [7]. SF manufacturing has been highly developed due to its various bioapplications and use in textiles. In addition, targeted changes to the processing of SF can yield entirely different products for different applications [36]. The structure of SF is enhanced through its processing from silk; SF proteins cross-link into β-sheets, which contributes to its superior strength [37]. When implanted into tissue, SF shows mild or no immunogenicity [38]. Similar to collagen, SF is degraded by proteases, although this occurs over longer periods of time and does not cause degradation of its surroundings [39]. The gradual process of SF degradation actually allows weight to slowly shift onto the endogenous ECM rather than leaving a structural weakness in its embedded tissue. SF biomaterials are ideal for wound treatment, and exploration into their applications in a wide range of therapeutic fields is warranted.

SF scaffolds are inexpensive, widely abundant, and contribute to optimal conditions for wound repair. Unlike many synthetic polymers, naturally derived SF scaffolds in the wound bed act as a biodegradable matrix that allows endogenous cells to cover and ultimately replace it with endogenous ECM. When processed, SF retains its porous, three-dimensional architecture and strength and can even be engineered to take on different nanoarchitectures [40]. Additionally, due to its porosity, SF allows for adequate water retention and gas exchange, which allows for a wound microenvironment conducive to proper healing [41].

### 3.1. SF-Composite Scaffolds

SF tissue scaffolds can be combined with other biomaterials, termed composite scaffolds, for a synergistic effect on tissue repair. These biomaterials can include growth factors, antimicrobial agents, therapeutics, and more. Composite scaffolds retain the strong mechanical properties of SF and provide additional properties which aid in wound healing. In theory, the dynamic ability of SF to be combined with other biomaterials allows it to treat specific healing deficiencies within a wider range of ailing wounds. Instead of a single treatment being applied to a broad range of wounds, additional targeted biomaterials can potentially help specific wounds heal.

Chitosan (CS), a biopolymer derived from shellfish, is one example of a biomaterial that has been integrated into SF tissue scaffolds [42,43]. The SF-CS-composite scaffolds adopt properties from each independent biomaterial. These scaffolds are biocompatible, antimicrobial, hemostatic, permeable, and mechanically strong [44]. Combining biomaterials with SF does not reduce the characteristics that make it an optimal tissue scaffold but rather imparts additional properties that enhance its overall effectiveness. Particularly, CS gives the tissue scaffold strong antimicrobial effects that allow it to heal without added inflammatory stress and infectious burden. Full-thickness rat wounds treated with SF-CS scaffolds healed effectively and without any long-lasting change to tissues [43]. Interestingly, upon treating newly inflicted wounds, little to no bleeding was observed, demonstrating the scaffold’s powerful hemostatic capabilities. The CS within the scaffolds formed a thick gel within the wound, which helped regulate hemostasis, nutrient delivery, and the microclimate of the wound. SF-CS-composite biomaterials showed increased mechanical strength by 130% compared with SF scaffolds alone due to increased hydrogen bonding interactions between the two biomaterials. The synergy observed in SF-CS scaffolds demonstrates the potential for other SF-composite scaffolds.

As another example of a composite scaffold, SF has been cross-linked with gelatin-derived biomaterials to improve wound healing. Gelatin is useful in tissue engineering because it does not exhibit an immune response, is biodegradable, and it is easy and cost effective to manufacture [45]. However, it is not an ideal scaffold on its own because it does not maintain its structure and can be very fragile. Additionally, while SF has the nanoarchitecture conducive to cell migration and proliferation, it lacks the natural peptides that specifically direct these activities. The cell-binding peptides in gelatin assisted 3T3 mouse fibroblast cells in populating an ECM mimic [46]. Another study cross-linked SF with gelatin-methacryloyl (GMC) and the nitric oxide (NO) donor molecule *S*-nitroso-*N*-acetylpenicillamine (SNAP) [47]. NO is a powerful antibacterial agent and can accelerate healing in chronic ulcers, namely those in diabetic wounds known to be NO-deficient [48]. In this case, SF was integrated into the previously studied GMC-SNAP hydrogel and improved its overall wound healing functions; SF enhanced NO-release mechanisms, the structural integrity of the scaffold, and protected the GMC from degradation in the wound [47]. SF-GMC tissue scaffolds also assisted fibroblast adhesion to the scaffold in the wound, resulting in accelerated healing. Another study used CNP-SNAP treatments and saw relatively increased antibacterial properties [49]. Given that SF has been used in conjunction with both of these biomaterials, further research into an SF-CNP-SNAP wound treatment is warranted.

The wound healing properties of SF can be greatly enhanced by other biomaterials. While already studied with common biomaterials such as collagen, gelatin, and chitosan, research into SF scaffolds composited with other materials has the potential to yield effective treatments. Specific composite-SF scaffolds can treat specific wound ailments. For example, SF-GMC tissue scaffolds are likely effective when treating wounds in which healing has been arrested in the inflammation phase, when risk of infection is highest. Alternatively, SF combined with a naturally occurring ECM component would likely help with the proliferation and/or maturation phases of wound healing, as cells more easily migrate onto the scaffold to allow for the rebuilding of the endogenous ECM. More research is needed on SF-composite scaffolds to elucidate new wound therapies because the potential treatments are promising.

### 3.2. Cellularized and Decellularized SF Scaffolds

Tissue scaffolds are bioengineered to mimic the ECM for a variety of reasons, but perhaps the most important of these is to provide a three-dimensional environment with physical properties conducive to cell migration, adhesion, and proliferation. The influx of cells into a wound is imperative for its repair and healing. SF tissue scaffolds serve as an ideal biomaterial for delivering multipotent mesenchymal stem cells (MSCs) into the wound area for improved healing. MSCs can aid wound healing by providing an ECM for adjacent cells to migrate into the wound, and by delivering pre-seeded cells to the wound for expedited tissue regeneration. Cells with multilineage differentiation capacity, such as MSCs, have been extensively shown to improve wound healing through their ability to renewably produce multiple cell lineages and release paracrine factors that can induce angiogenesis and cell proliferation [50,51]. MSCs, which are derived from multiple different tissues including bone marrow (BM), Wharton’s jelly, and adipose tissue, play important roles in wound healing. MSCs are responsible for contributing fibroblasts, keratinocytes, cytokines, and ECM proteins to the wound, and thus have great therapeutic applications to enhance the wound healing process [52]. For example, it has been shown that BM-MSCs treat wounds more effectively than neonatal dermal fibroblasts, despite the latter’s role in promoting cutaneous growth and preventing scarring in fetal tissue [53,54,55].

Despite their huge potential, transplanted MSCs alone often have poor cell viability and are prone to losing multipotency due to various environmental sensitivities, markedly decreasing their effectiveness as a therapeutic agent in wounds [56,57]. Seeding MSCs and other multipotent stem cells ion SF tissue scaffolds solves this issue; SF scaffolds have been shown to maintain MSC viability and differentiation potential in wounds [20].

Given the success that acellular SF tissue scaffolds have had in creating a microenvironment and nanostructure for endogenous cells to more efficiently regenerate damaged tissue, it comes as no surprise that these scaffolds are also ideal for supporting MSCs and the delicate environment they require to properly function. The complex and variable conformations that make up SF tissue scaffolds closely resemble the non-uniform ECM seen in nature. This mimicry provides a suitable nanostructure for MSCs to adhere, secrete paracrine factors, and to differentiate into the mature progeny cells that aid in wound repair [58]. The physical properties of the tissue scaffold into which stem cells are seeded can determine much of their behavior and phenotype once implanted. Pore size, porosity, stiffness, elasticity, and topography are among the most important physical factors of a tissue scaffold that influence a cell’s behavior [58], and seeded stem cells are particularly sensitive to these conditions. For example, whether a stem cell remains pluripotent or enters senescence can depend entirely on the stiffness of the tissue scaffold; murine stem cells maintained pluripotency on a soft substrate (6 kPa), which was then entirely reversed when seeded on a stiff substrate (47 kPa) [59]. In SF tissue scaffolds, dermal fibroblasts with a pore size of approximately 200 µm proliferate more rapidly than those with a pore size of 75 µm [60]. A major benefit of SF tissue scaffolds is their modifiable, three-dimensional conformation. The nanoarchitecture can be manipulated through different SF manufacturing techniques, allowing scaffolds to be specifically tailored to different therapeutic approaches, depending on the deficiencies of a given wound [61]. The dynamic physical characteristics of SF tissue scaffolds are optimal for MSC adhesion, differentiation, and paracrine factor secretion.

The most common source of MSCs for wound treatments are those derived from adipose tissue (Ad) as they are easily isolated and the cell population can be readily expanded in vitro. Ad-MSCs frequently have detectable CD34 expression, indicating their ability to induce angiogenesis. It has been proposed that Ad-MSCs lie dormant until they are recruited to a wound to induce vascularization and regeneration. Unfortunately, they are not prolific enough in endogenous tissue to adequately repair severe wounds, highlighting the opportunity for an Ad-MSC delivery system into the wound. SF tissue scaffolds, frequently combined with CS for its antimicrobial properties, have been shown to be effective at maintaining Ad-MSC viability and multipotency in vivo as they populate a wound with mature progeny cells. In one study, mice were given full-thickness wounds and treated with unseeded SF-CS scaffolds, Ad-MSC-seeded SF-CS scaffolds, or no tissue scaffold [62]. The wounds treated with Ad-MSCs showed enhanced recovery when compared with unseeded SF-CS scaffolds and untreated controls. Multiple cell lineages were observed, indicating that MSCs in the wound were able to differentiate into the appropriate cells for tissue regeneration.

Another study evaluated the in vivo effects of Ad-MSC-seeded SF scaffolds on murine wounds. Additionally, this study examined the effect of Ad-MSC-decellularized SF scaffolds [63]. Ad-MSCs were cultured on SF-scaffold patches in vitro. The patches were subsequently decellularized and, along with the Ad-MSC-seeded patches and a negative control, implanted into wounds to study the effects of the non-cellular factors that MSCs secrete. Interestingly, the decellularized SF scaffolds were nearly as effective as the seeded scaffolds. Both showed enhanced repair compared to the negative control. These results suggest that the porosity of SF tissue scaffolds, similar to the ECM, help to serve as a depository for MSC-secreted growth factors and other stimuli that signal for surrounding cells to begin—or in many chronic wounds, resume—tissue regeneration. While implantation of seeded Ad-MSCs was the most effective treatment, this study highlights the potential of SF scaffolds because they not only provide cells with the proper physical microenvironment to adhere and grow but also inherently contain the chemical properties required to bind the MSC-secreted factors that attract the appropriate endogenous cells to the wound for repair. Furthermore, it shows that while implanted exogenous MSCs are effective modulators of regeneration, they are not a necessity for improved wound healing, which potentially avoids immunogenic responses in sensitive individuals.

Bone marrow is another common source of MSCs for wound treatment, and it has many similarities to Ad-MSCs [64]. BM-MSCs are isolated from aspirated bone marrow. These BM-MSCs have been reported to enhance wound healing through expression of keratin and the formation of both vascular and glandular structures [54,65]. Interestingly, murine wounds treated with BM-MSCs have shown significantly better wound recovery when compared to those treated with neonatal dermal fibroblasts [54]. In one study, BM-MSCs were evaluated to determine their compatibility and effectiveness when seeded on SF-CS tissue scaffolds for wound treatment [58]. It was determined that BM-MSCs maintain multipotency, increase vascularization, assist in cell adherence, and allow motility into and throughout the wound in this setting. Notably, BM-MSCs differentiated into both osteogenic and adipogenic cell lineages, both of which are essential for tissue regeneration. The presence of lipid droplets and mineralization nodules indicated the differentiation of BM-MSCs into adipogenic and osteogenic cells lines, respectively. Additionally, treated wounds showed no immunogenic response, demonstrating the biocompatibility of both SF-CS scaffolds and BM-MSCs [7].

Wharton’s jelly (WJ) is the gelatinous substance in the umbilical cord, and it contains mucoid connective tissue and multiple different cell types, including MSCs. WJ-MSCs are ideal for treating dermal wounds as they have also been shown to undergo both adipogenic and osteogenic differentiation [66]. WJ-MSCs are cost effective and easily obtained from donated umbilical cord tissue [67]. Compared to BM- and Ad-MSCs, WJ-MSCs have the highest proliferation capacity and show little to no evidence of immunogenicity or senescence, which suggests that they would be the preferred treatment for quickly achieving wound closure by populating a wound with progeny cells. Comparable to BM- and Ad-MSCs, WJ-MSCs maintain viability and proliferative capacity when seeded on SF tissue scaffolds [66]. In one study, SF scaffolds seeded with human WJ-MSCs were implanted into murine full-thickness dermal wounds [20]. Upon seeding, the SF scaffolds maintained their original structure. WJ-MSCs took on a flattened fibroblastic morphology and moved throughout the SF scaffold’s three-dimensional pores to form a monolayer over the scaffold. The seeded cells integrated with the SF fibers, using them similarly to an endogenous ECM. WJ-MSCs secreted essential collagen into the wound, indicating the regeneration of the natural ECM as the SF scaffold degraded over time. Wounds treated with WJ-MSC SF scaffolds also showed significantly higher vascularized areas compared to untreated controls. Additionally, CD90-positive cells, which indicate the presence of MSCs, were observed within the granulation tissue, confirming that WJ-MSCs differentiated into epithelial cells within the wound [20].

MSCs can be derived from a number of different allogenic and autologous sources. While cell behavior and morphology from specific sources vary slightly, it has been shown that multiple sources of MSCs can be seeded on SF tissue scaffolds for improved wound healing. Ad-, BM-, and WJ-MSCs have all been shown to maintain vitality, multipotency, proliferative capacity, and motility when seeded on a tissue scaffold and implanted in a wound. MSCs adhere to and integrate with SF fibers, allowing for integration into the wound. MSCs also secrete collagen, which is pertinent for replacing the biodegradable tissue scaffold with a natural ECM. Additionally, vascularization was increased in wounds treated with seeded SF scaffolds. Importantly, decellularized SF scaffolds also improved wound healing, indicating their ability to bind and deliver paracrine factors and other MSC-secreted growth factors. Ultimately, SF tissue scaffolds have been shown to have great success in MSC delivery into wounds.

## 4. SF Solution

It is well established that the skin of patients with diabetes is deficient in critical wound healing mechanisms and properties, including inferior mechanical properties compared with the wounded skin of healthy individuals. Upon investigation of the strength of the skin of patients with diabetes, our group was the first to report significant biomechanical weaknesses at baseline [68]. The skin of patients with diabetes showed an average maximum stress of 0.62 MPa, while healthy skin was 2.10 MPa, on average. Additionally, modulus elasticity testing recorded the skin of patients with diabetes at 1.82 MPa, compared with 7.72 MPa for healthy skin. These weaknesses are a key component in the increased risk of skin injury in individuals with diabetes, which exacerbates their already impaired wound healing. Strengthening the skin of patients with diabetes can therefore help prevent injury and improve healing.

An SF solution can be made by completely dissolving the SF protein in an aqueous solution [69]. The highly tensile SF has a strength-to-density ratio greater than steel, making it an ideal biomaterial for strengthening skin [19].

### 4.1. Nanosilk

Nanosilk is an SF-based solution which our group applied to human diabetic skin samples to improve its biochemical strength [14,70]. Skin samples were treated with a topical application of either 7% nanosilk or a control solution. We found significant increases in both the maximum load and modulus of diabetic skin samples treated with nanosilk; the maximum load of the diabetic skin treated with nanosilk increased by 23% compared with untreated controls, while modulus increased by 35%. No difference in skin elasticity between the treated and untreated groups was observed (Figure 1). The increased tensile strength of 51 MPa from 42 MPa indicates the powerful ability of SF to reinforce skin, even when applied topically in a solution. When topical nanosilk was applied as a cream, healthy facial skin was also shown to be more resilient [71]. Wound prevention is often overlooked when investigating solutions for impaired wound healing. Our results suggest that there may be some benefit to regular application of nanosilk to both the unwounded and wounded skin of individuals with diabetes. Strengthening at-risk skin may decrease the incidence of chronic ulcers, thus alleviating some burden on the healthcare system and lowering the overall cost of treatment.

### 4.2. SF Solution Delivery System

MicroRNAs (MiRs) are small, non-coding RNAs that have regulatory oversight of many cellular processes through translational inhibition via the RNA interference pathway [72]. MiRs are often dysregulated in pathological tissues such as the wounds of individuals with diabetes, leading to aberrant expression of target genes and the development of chronic ulcers. MicroRNA146a (miR146a) is a powerful regulator of the inflammation response mediated by the nuclear factor kappa B (NFκB) pathway in wounds. In the wounds of individuals with diabetes, miR146a expression is dysfunctionally downregulated, leading to chronic ulcers that are unable to progress beyond the inflammatory phase of healing [73]. Unfortunately, unmodified miRs are not a viable option for treating wounds as they are quickly degraded and show highly limited cellular uptake [74]. For efficient delivery into wounds, our group conjugated miR146a molecules to vacancy-engineered cerium oxide nanoparticles (CNP) [75]. CNP serves as a novel delivery mechanism for miRs into wounds by stabilizing their negative charge. Additionally, CNPs alone have strong free-radical-scavenging properties and have been shown to relieve oxidative stress [76]. We tested the effect of CNP-miR146a on wound healing by applying it intradermally to 8 mm full-thickness excisional wounds in diabetic mice. Treated wounds were significantly smaller after 14 days and showed higher angiogenic- and lower pro-inflammatory-gene expression, indicating a healthier wound environment. While intradermal delivery of CNP-miR146a was effective, it is invasive and when translated to human medicine, injections would likely need to be performed by clinicians.

To investigate wound treatments that are more accessible and less invasive, we used topical nanosilk to deliver CNP-miR146a to wounds. Full-thickness murine wounds were treated with a nanosilk solution containing CNP-miR146a molecules, nanosilk only, or a control. Wounds were monitored until full closure, then harvested for biochemical and biomechanical analysis. Wounds treated with nanosilk containing CNP-miR146a healed significantly faster than controls. Real-time quantitative polymerase chain reaction showed significantly increased expression of the pro-fibrotic genes Col1α2 and TGF β-1, significantly decreased IL-6 and IL-8 pro-inflammatory gene expression, and higher collagen levels in treated wounds. This dual-effect treatment corrects the dysregulated inflammatory response in the wound by scavenging free radicals and downregulating the NFκB pathway while also providing tensile strength to the weakened skin [14,70].

SF has been shown to effectively deliver miRs and nanoparticles to the wound site. While miR-146a is pro-inflammatory, other miRs could have a beneficial impact on a healing wound. For example, if a wound is treated with an miR that has a pro-fibrotic effect, improved wound healing is possible. As we uncover the roles of more miRs, it is reasonable to infer that nanosilk could effectively deliver them to wounds.

## 5. SF-Based Sensors

Biosensors are devices that interact with bioactive molecules to provide quantifiable feedback from complex biological pathways, reactions, and mechanisms. These devices are highly sensitive and are instrumental in elucidating the status of a given tissue in terms of inflammation status, glucose levels, reactive oxygen species (ROS), and other bioindicators. The feedback provided by biosensors can be applied to drug discovery, pathology, diagnostic and prognostic tools, and even food safety [77]. The early development of biosensors was largely focused on electrochemical interactions using reactive electrodes, such as gold or platinum, as indicators of glucose concentration [78]. These sensors used electrodes to detect byproducts of enzymatic reactions, such as glucose oxidase. A quantifiable indicator, such as color or fluorescence, is measured for analysis. Since then, biosensors have been used in a wide variety of detection applications, including immune response, DNA hybridization, and sweat [79]. Modern biosensors have been composed of biomaterials such as SF, CS, silica, and cellulose. Organic biosensors often carry the added benefit of high biocompatibility, controlled degradation, and low immunogenicity. These sensors serve a very important purpose in quantifying the biometrics that can be monitored or targeted for treatment.

The ideal biosensor responds to the specific conditions it is designed to measure. A simplified example of this mechanism has been shown in glucose monitoring. As glucose is oxidized by glucose oxidase, hydrogen peroxide and atmospheric oxygen are synthesized as byproducts. These byproducts interact with a platinum anode, creating an electrochemical signal that can be detected and quantified, depending on the amount of byproduct present [80]. While the specific mechanisms vary, the purpose of biosensors largely remains constant: to react with a target molecule as a means of producing a measurable signal. In this review, biosensors derived from SF and their use in wound healing are discussed.

Enzyme immobilization is often necessary for a biosensor to function reliably. Enzymes that produce the byproducts being analyzed are trapped between or within a membrane. Here, they carry out their cellular function and provide molecules for the biosensor to detect [80]. Immobilization is usually carried out by inorganic molecules. Decades ago, it was shown that SF can effectively immobilize enzymes and serve as an organic biosensor [81]. The porous nanoarchitecture of SF allows it to absorb different enzymes while preserving its functionality. SF has also been shown to immobilize enzymes with covalent bonds. For this reason, SF has been used as a biosensor to determine glucose concentration by immobilizing glucose oxidase. SF also immobilized peroxidase in a similar experiment, indicating its potential to immobilize a broad variety of enzymes as well as the potential biosensing applications that are possible [81]. CS-SF biosensors have recently been shown to have incredible sensitivity when immobilizing phytase, highlighting its continued focus as a sensing biomaterial over the decades [82].

The ability of SF membranes to immobilize enzymes such as uricase, phytase, and glucose oxidase has made it a popular choice as a biosensing material. Systemic biosensors are important when a disease affects more than one tissue. Accurate monitoring of these conditions is integral to their treatment and cure. For example, gout and Lesch-Nyhan syndrome both involve the dysregulation of purine degradation, leading to abnormal uric acid levels [83,84]. SF membranes have been shown to effectively immobilize uricase, the enzyme responsible for the latter steps in purine metabolism [21]. These SF-based biosensors are able to detect uric acid in bodily fluid, allowing for faster and more specific treatment in affected individuals. Another study incorporated SF into a biosensor gel that gave readouts of body motion and temperature, allowing for continuous monitoring of patients’ internal temperature [85]. This sensor was non-invasive and could relieve clinicians of the time spent taking and recording temperatures; additionally, it could indicate rapid body movements such as seizures and alert the appropriate personnel.

More recently, different SF biosensors have successfully been applied to wound healing. Chronic wounds differ from the normal wound environment and healthy skin in a number of ways. Upregulation of pro-inflammatory proteins and increased ROS leading to lower pH are both indicative of chronic wounds. Accurate detection and quantification are important for choosing the proper treatment. Current ROS detection is accurate but not in real-time and often cannot detect low ROS levels [86]. To our knowledge, our group is the first to develop a colorimetric biosensor that detects the presence of ROS within wounds [87]. The SF biosensor turns from white to red in the wound, visibly indicating a wound that may be chronic (Figure 2). Hydrogen peroxide, one of the most common ROS, reacts with Amplex red, a fluorogenic probe, to produce a red color [22]. As hydrogen peroxide reacts with the biosensor, the intensity of the pink-to-red color provides a visible indication of ROS levels in the wound (Figure 3). Understanding ROS levels expedites the process of treatment and can elucidate which medications are needed and offer a prognosis for wound healing. Another SF biosensor—a membrane consisting of SF and nanodiamonds—can be implanted into wound beds for temperature monitoring; higher temperatures can indicate infection or increased inflammation [88]. These biosensors also have antibacterial properties, helping to prevent infection as well.

SF biosensors have been used for decades, and their continued use indicates strong potential for future SF biosensing applications. SF biosensors have shown the ability to immobilize enzymes and react with ROS in wounds. The degradability of SF allows it to be implanted into areas that would otherwise need to be re-accessed for removal. By visually indicating the presence of ROS, potentially invasive or even inconclusive tests are avoided. Additional biosensors that detect different biomarkers typical of chronic wounds should be explored.

## 6. SF Nanoparticles

Over the last decade, the effects of nanoparticles on dermal wounds have been extensively studied. Nanoparticles are 100 nanometers (nm) or smaller, and often have unusual and unique chemical and physical properties independent of their macro forms. In the wound environment, nanoparticles can aid the healing process by improving angiogenesis, decreasing inflammation, conferring antimicrobial effects, regulating gene expression, and altering the ECM [89]. Nanoparticles also enable targeted delivery of therapeutics, metals, exogenous RNA, and other organic and inorganic wound treatments; they are most often used in the form of polymeric nanoparticles, nanotubes, micelles, liposomes, nanometals, drug conjugates, and protein carriers.

Traditional medication-delivery systems, such as oral ingestion and intravenous or intramuscular injections, often limit the delivery of novel therapeutics. These systems are frequently limited by high degradation, low bioavailability, and poor targeting of diseased tissues [90]. Ideal drug delivery systems should be non-toxic, biocompatible, and allow for controlled dosing and targeted release. Nanoparticles have become a primary focus in the development of drug delivery systems. Various synthetic and natural nanoparticles have been used to successfully modulate the localization, timing, and uptake of growth factors, proteins, and drugs in tissues. While numerous synthetic biomaterials have shown success as slow-release drug delivery systems, they are not ideal for carrying all novel therapeutics due to instability and toxic synthesis processes [91,92]. Organic solvents, surfactants, and cross-linking agents are often used to make synthetic polymers, which can lead to adverse reactions in vivo. Nanoparticles from natural polymers, like SF, stand out as ideal drug delivery systems because of their modifiable nanostructures, biocompatibility, and customizable degradation. However, natural polymers’ variability in structure and drug release could potentially limit their applications in select cases [10].

The behavior of a given nanoparticle is determined by both its structural and chemical properties. Particle size is the most influential property as it determines targeting ability, cell uptake, and drug release. While specific rates vary by material, smaller nanoparticles have been shown to have higher cell uptake than their larger microparticle counterparts [93]. Larger particles are associated with slower drug release as the encapsulated drug is further from the nanoparticle’s surface; conversely, smaller particles exhibit faster release as the drug is closer to the nanoparticle’s surface [94]. Additionally, smaller particles diffuse through tissue more easily than larger particles, leading to wider drug distribution. Shape is also an important factor in how a nanoparticle behaves. One study demonstrated that spherical nanoparticles had much more efficient uptake than rod-shaped nanoparticles of the same material [95]. Chemical properties such as hydrophobicity and particle charge can determine the target cell’s fate. Hydrophobic particles are more likely to attract phagocytes, leading to targeted cell death [96].

SF is presented here as a nanoparticle for controlled delivery of bioactive therapeutics as it is highly dynamic and can be manipulated for various biomedical applications through well-characterized, non-toxic processing methods [10,36,40,58,97]. The molecular and physical properties of SF allow for highly customizable particle size, ranging from about 10 nm to over 100 nm [24]. SF nanoparticles have also been made into different shapes as well as combined with other biomaterials, depending on their intended applications [25]. SF is an FDA-approved therapeutic biomaterial and has been widely used in sutures, wound dressings, and tissue scaffolds because of its biocompatibility and non-cytotoxicity [98]. It has been shown to efficiently deliver growth factors, proteins, and other novel therapeutics to wounds and other tissues.

The use of SF nanoparticles as a delivery system for novel therapeutics has grown in popularity due to the non-toxic processes used to prepare the drug delivery systems; SF nanoparticles can be prepared without organic solvents or other cytotoxic chemicals through a variety of processes including milling, electrospraying, freezing, and desolvation [99]. The nanoparticles can then be loaded with drugs and other therapeutics by the simple process of adsorption, which is enabled by SF’s porosity [25,100,101,102]. For subsequent drug delivery, nanoparticles can be engineered for different release behaviors. Importantly, SF nanoparticles are small enough to easily penetrate tissues, thus increasing the efficiency of drug uptake [24]. After drug release, the high biocompatibility and low immunogenicity of SF nanoparticles leads to either natural degradation or passive clearance, without adverse effects [10].

Topical delivery of SF nanoparticles conjugated to therapeutic agents can effectively improve wound healing. Aerosolized SF nanoparticles were used to topically deliver *Avicennia marina* extract and neomycin into full-thickness rat wounds to successfully enhance healing [103]. *Avicennia marina* is a wooded plant, the extract of which has been found to have anti-inflammatory, antioxidant, and antimicrobial effects [104]. The extract has also been shown to stimulate the proliferation of fibroblasts and induce epithelization [103]. Neomycin is a common antibiotic used to prevent infection in a variety of tissues, including cutaneous wounds [105]. On the first day after treatment, SF nanoparticles released 49% and 68% of the neomycin and *Avicennia marina* extract, respectively. This was followed by a gradual release of the remaining treatments over the next 24 days. The initial release could be beneficial for immediately killing any bacteria in the wound bed and stimulating the proliferation of fibroblasts, while the subsequent prolonged release could continue to promote a healthy wound bed throughout the healing process. Wounds treated with SF-loaded nanoparticles healed within 15 days, while negative control wounds remained open in the same timeframe [103]. The success of topical SF nanoparticles as a drug delivery mechanism suggests that the nanoparticles could be used dynamically with various drug combinations to treat a number of common wound ailments.

Dual-drug delivery involves multiple drugs being delivered and released through one system. These systems aim to selectively deliver drugs to target tissues for a synergistic effect while also controlling their release [106,107]. Hydrogels are a popular wound treatment and drug delivery system because they can carry multiple therapeutics within their polymeric networks and can enact their release with the appropriate stimulus, such as heat or pH changes [99]. However, the variable network structure of the biomaterials used to synthesize hydrogels can lead to difficulty modulating drug release in certain situations [108]. SF nanoparticles provide a promising solution for this as they can be loaded with bioactive materials and cross-linked into the polymeric network of both synthetic and natural hydrogels for regulated release of novel therapeutics such as genes, proteins, and growth factors [109].

Bacterial infection is a leading cause of chronic wound development and can be life threatening if left untreated [3]. Drug-resistant bacteria such as methicillin-resistant *Staphylococcus aureus* (MRSA) have become increasingly difficult to treat with traditional therapeutics [110]. Chronic wounds are well documented to have reduced epidermal growth factor (EGF) expression, slowing tissue regeneration [111]; however, EGF therapeutics have historically been unsuccessful due to their instability in the harsh conditions of a wound bed. Novel SF-nanoparticle-based drug delivery has recently been shown to enhance healing in chronic wounds and kill pathogenic bacteria through the controlled local release of antibiotics and growth factors [112]. Alginate-SF nanoparticles loaded with vancomycin are cross-linked to poly(*N*-isopropylacrylamide) (PNIPAM) hydrogels containing EGF. Alginate-SF nanoparticles release vancomycin, a powerful antibiotic against MRSA, at a pH-controlled rate in the presence of the alkaline conditions of chronic wounds [113]. EGF is steadily released and stabilized by the PNIPAM hydrogel. After treatment, both EGF and vancomycin were released into the wound for the entire 20-day study duration. Importantly, the bioactivity of the therapeutic agents was maintained at 80%, demonstrating the stabilizing ability of the SF-based delivery system. Treated wounds showed improved wound healing at 21 days—91% closure in treated wounds compared with 42% closure in controls—higher growth factor expression, and reduced bacterial infection, which demonstrated the potential of SF nanoparticles as a means of novel drug delivery in wounds.

While a wide range of therapeutics has been delivered in vitro and to pathogenic tissue by SF nanoparticles, research into SF nanoparticles as a wound healing treatment is scarce. However, these therapeutic applications can still provide insight into the underlying release mechanisms and interactions of SF nanoparticle-based drug delivery and how they could be applied in cutaneous wounds. For example, an in vitro study examined the prospect of sustained growth-factor release by SF nanoparticles as a novel therapeutic [114]. Vascular endothelial growth factor (VEGF) conjugated to SF nanoparticles was successfully released into cells over a three-week period. The nanoparticles delivered stable VEGF at a controlled, rapid pace for the first five days before release slowed, suggesting this delivery system could be used to supplement growth-factor-deficient wounds before slowing release to maintain normal levels. However, the in vivo stability of protein therapeutics is much more difficult to achieve, implying that more research needs to be conducted.

Model drugs can also be used to form an understanding of how an SF nanoparticle interacts with different drugs and therapeutics. For example, a study was conducted on the controlled release of model drugs by SF-nanoparticle-SF-hydrogel delivery systems [115]. Fluorescent dyes were conjugated to SF nanoparticles, allowing the characterization of release behavior in the presence of mesenchymal stem cells. Three dyes were loaded onto SF nanoparticles: Rhodamine B (RhB) and Texas Red (TR), which are both hydrophilic, and fluorescein isothiocyanate (FITC), which is hydrophobic. The release behavior, encapsulation efficiencies, cumulative release, and conjugate structure were observed; hydrophobicity and size ultimately determined how each dye behaved as smaller RhB and TR followed a nearly identical pattern in contrast with the much larger FITC [115]. RhB and TR were quickly released from within the nanoparticles, while the hydrophobic interactions between FITC and SF were reported to slow proteolytic degradation and, consequently, FITC release. The difference in degradation rates alone suggests that the combination of the hydrophilic and hydrophobic biomaterials loaded onto SF nanoparticles can provide dual-drug delivery with independent release behaviors.

While the behavior of therapeutic nanoparticles is clearly multifactorial and sound conclusions cannot be made from in vitro therapeutic behavior, new research avenues could be elucidated. SF nanoparticles should be examined as a novel, in vivo drug-delivery system in wounds.

## 7. SF Hydrogels

Hydrogels are three-dimensional networks of cross-linked polymers [116]. They also require a high proportion of water. When applied to wounds, hydrogels can facilitate drug delivery through their networks of polymers. Hydrogels can also create an exudate for wounds to absorb, keeping them moist for proper wound healing conditions. While hydrogels have been made from many different types of polymers, natural polymers have the most appeal for treating chronic wounds due to their biocompatibility and low toxicity. Of the natural polymers that have been used, SF has emerged as a promising candidate due to its dynamic structure and controlled biodegradability.

SF hydrogels are easy to manufacture. The pH of the SF solution is simply lowered with an acidic solution, and gelation is induced. As a drug carrier, this can be limiting as a lower pH may not be suitable for all drugs [10]. Other methods, such as sonication, have been applied to the gelation of SF solutions [117]. These methods are more complex and less practical for the quick and widespread production of treatments; however, they do provide an alternative to pH-restricted drug delivery. Additionally, SF hydrogels do not possess the same mechanical strength of their SF counterparts [18]. For these reasons, SF hydrogels are somewhat limited in application. Nevertheless, they have still been used to significantly improve wound healing.

Due to the limitations of SF hydrogels, other biomaterials are often employed to help increase functionality. One such study combined tannic acid, chitosan, and SF into a hydrogel [44]. The addition of tannic acid increased mechanical strength by up to five-fold by cross-linking into the existing SF polymeric network. Tannic acid also conferred antimicrobial properties to the hydrogel. When applied to full-thickness murine wounds, wound closure was achieved significantly faster than in controls. The authors speculated the addition of tannic acid may have improved free-radical-scavenging capacity.

Despite the apparent limitations of SF hydrogels, researchers have found innovative applications for the hydrogels in wound treatment. Another study found that SF hydrogels can be made to undergo gelation at the local treatment site [118]. The use of silk fibroin from both *B. mori* and *Antheraea assama* led to the self-assembly of β-sheets and subsequent cross-linking. Wound healing was improved when compared with collagen controls.

While SF hydrogels may not be the optimal wound treatment, they have still been shown to improve healing overall. Unfortunately, hydrogels are hindered by limitations in drug delivery and manufacturing. This suggests that more focus should be put into other areas of SF therapeutics as they relate to wound healing. However, it is not unreasonable to research drugs that are stable in a low-pH environment for use in SF hydrogels.

## 8. Conclusions

Wound healing is a highly delicate process that involves orchestrating the activity of the ECM, various cell types, growth factors, and other proteins. Impaired healing in wounds can lead to a variety of health problems, including sepsis, necrosis, and death. As the rate of individuals who develop chronic ulcers continues to rise, the need to find new wound therapeutics in various clinical settings only becomes more urgent. SF-based biomaterials have been presented as the ideal wound treatment as they are cost effective, dynamic, non-cytotoxic, biodegradable, and highly biocompatible. The natural polymer’s customizable nanostructures and tunable degradation have led to numerous wound applications, namely SF tissue scaffolds and nanoparticles. SF tissue scaffolds serve as an ECM mimic, allowing cells to adhere and proliferate within the wound. SF nanoparticles can modulate novel drug-delivery systems for controlled release of therapeutics, such as CNP-miR146a. Continued research into the delivery of drugs and growth factors to the wound will help elucidate the extent to which SF can aid wound healing. Additionally, biosensors, such as the SF mat that indicates ROS within wounds, are instrumental in both the prognosis and diagnosis of chronic wounds.

## Figures and Tables

**Figure 1 pharmaceutics-14-00651-f001:**
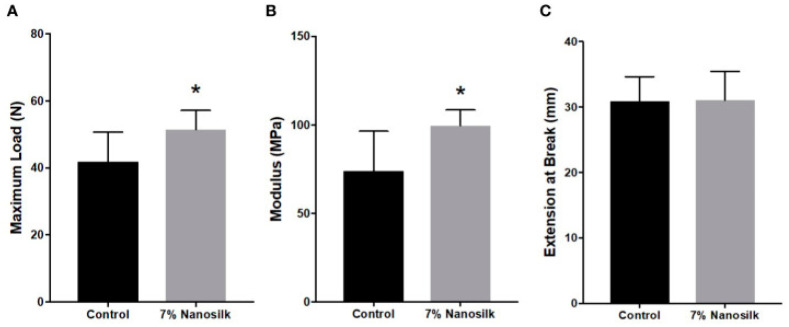
Mechanical strength of samples treated with 7% nanosilk or controls: (**A**) maximum load, (**B**) modulus, and (**C**) elasticity. Statistical significance (*p* < 0.05) is indicated by *. Standard acknowledgment: Reprinted with permission from Ref. [14]. 2020 Frontiers in Immunobiology.

**Figure 2 pharmaceutics-14-00651-f002:**
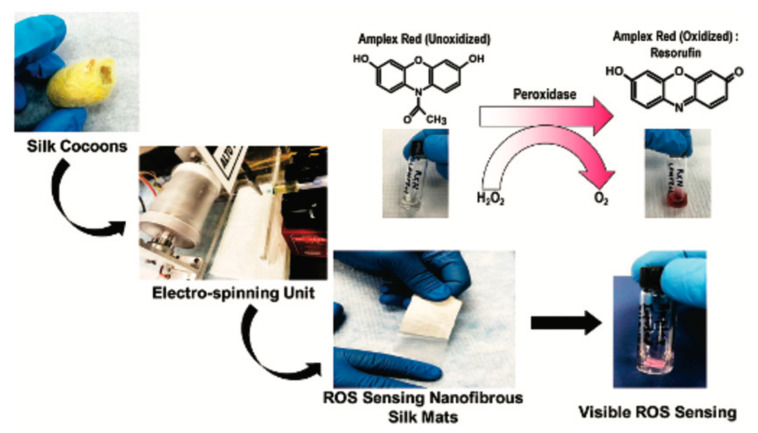
Colorimetric ROS detection. Reprinted with permission from Ref. [87]. 2020 Biomaterials Science.

**Figure 3 pharmaceutics-14-00651-f003:**
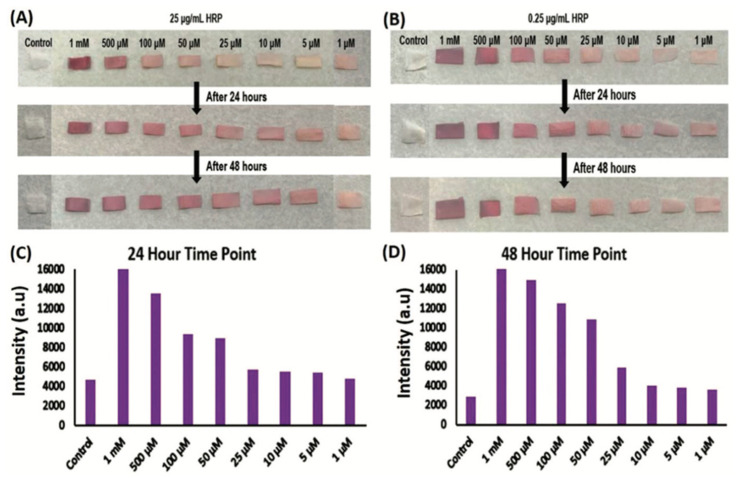
SF mats and control mats with Amplex red after treatment with known concentrations of HRP and hydrogen peroxide. (**A**) Fixed HRP concentration of 25 µg mL^−1^ and differing H_2_O_2_ concentrations at 24 and 48 h time points. (**B**) Fixed HRP concentration of 0.25 µg mL^−1^ and differing H_2_O_2_ concentrations at 24 and 48 h time points. (**C**) Colorimetric quantitation of intensity at 24 h time point. (**D**) Colorimetric quantitation of intensity at 24 h time point. Reprinted with permission from Ref. [87]. 2020 Biomaterials Science.

**Table 1 pharmaceutics-14-00651-t001:** Advantages and disadvantages of SF.

SF Biomaterial	Advantages	Disadvantages	Applications	References
tissue scaffolds	ECM mimic dynamic properties biomechanical strength	invasive treatment	tissue repair strengthens skin	[7,19,20]
solutions	topical application biomechanical strength	decreased solubility	drug delivery strengthens skin wound repair	[9,14]
biosensors	biomarker detection	minimal therapeutic value	wound monitoring	[21,22]
nanoparticles	customizable size short-term drug release	degrades over time unsuitable for long-term release	drug delivery	[23,24,25]
hydrogels	efficient drug delivery	swelling decreased mechanical strength	wound healing drug delivery	[11,26,27]

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
