# Peer review of "Silk Fibroin-Based Therapeutics for Impaired Wound Healing"

_pharmaceutics, 2022, doi:10.3390/pharmaceutics14030651_

Round 1
Reviewer 1 Report
This paper reviews the recent research related to the application of silk fibroin based therapeutics in wound healing. It discusses the different biomaterial applications of silk fibroin based nanotechnology has in aiding the wound healing process.
There are some concerns that need to be addressed before considering publication.
- The authors are advised to take the help of professional editing service to improve the language used within the text because there are a lot of grammatical mistakes, syntax errors, improper use of punctuation, and improper construction of sentences. Below are some examples where revision is recommended from the abstract section.
- In the abstract – use the term “treat” or “improve” instead of “correct”; similarly use the term “improving” instead of “correcting”
- Remove the following from the abstract “Our group was the first to demonstrate that silk fibroin increases the tensile strength of diabetic skin and reduces the risk of cutaneous injury” and add it into the introduction section with cited references
- Please write it “This review focuses on the different biomaterial applications of silk fibroin………” instead “This review focuses on the different biomaterial applications silk fibroin”
- Please revise the following “The applications of these biomaterials include nanoparticles, biosensors, tissue scaffolds, wound dressings, and novel drug delivery systems.” Nanoparticles, scaffolds, dressing materials are different forms of the Silk Fibroin biomaterial's, it does not come under the category of applications.
- I recommend rewriting the abstract, please provides a brief breakdown of the article to help readers to understand how the paper is useful to their research
- Figure 1, 2 and 3 is not so clear and a little blurry to understand. For readers, the quality of the pictures and the clarity of the text must be improved.
- Introduce the separate section which briefly explains the physicochemical properties of silk fibroin. Readers would expect in depth discussion on the therapeutic aspect of the common concerns encountered while using the silk fibroin such as solubility, bioavailability, chemical stability, sustained release properties and what type of nanocarriers have been efficiently applied to address the challenges.
- Discuss the stability issues of silk fibroin in the wound environment.
- Please write the biological properties and clinical application of silk fibroin
- As natural compounds (Silk fibroin) has an important contribution in this review, their chemical structure and how they interact with nanocarriers should be discussed from the literature.
- Nanogels can also be considered as another silk fibroin based nanocarrier for wound healing. However they were not considered in this review, I recommend discussing it.
- In conclusion part, specify the future potential research direction, the major gaps you find from the recent research. Please include the future perspectives of the work in Section 6 and revise the title “Future perspectives and Conclusions” instead of “Conclusions”.
- Please add a new Table and explain the various applications of silk fibroins and their limitations of each research.
- In page 13 - Please carefully check the Author contribution, funding, acknowledgements, conflicts of interest sections and write it appropriately.
Author Response
Please see attached response

Reviewer 2 Report
This manuscript summarized the Silk Fibroin-based therapeutics for impaired wound healing. The comprehension of silk fibroin-based nanotechnology would inspire the innovation of new cutaneous wound healing and tissue engineering applications. This could be interesting for the audience of Pharmaceutics. In addition, the contribution is of value for those colleagues working in the field of wound healing products. Generally, this is a great manuscript. It is well structured and technically sound and can contribute to the use of silk fibroin. A very positive point in favor of this manuscript is the constant presence of studies by the team itself. Thus, indicating the expertise that review authors must have. A plagiarism check was performed, and a nominal value (1%) was reported (referring to small fractions, 8-9 words, of sentences and figure 2 legend). As a person from a non-English-speaking country, I would prefer not to evaluate the quality of grammar and stylistics of English in this paper.
There are some notes and changes to be considered.
- The TITLE is enlightening and pertinent, and the ABSTRACT is well-structured with the expected content. However, I do not see the need to refer COVID-19 on the abstract. The REFERENCES are relevant, but, for the most part, they are not as recent as I expected.
- "29 million people in America" - a percentage of the population would be more comprehensive;
- Once again, I do not see the need to refer to COVID-19. It is not explored in the rest of the paper nor supported by the references (3,4).
- "SF is derived from cocoons of Bombyx mori. " Gives a wrong idea. SF fibers are obtained from diverse sources. Please clarify.
- The focus of the review will be on silk, OK. However, other promising biomaterials under recent investigations should also be briefly mentioned in the introduction.
- Please consider the possibility of creating an illustration summarizing the manuscript sections (2-5), reflecting the literature review carried out. A well-organized table could also be attractive. One of these approaches would be helpful for the reader to rapidly understand the strategies included in the manuscript as a whole.
- Page 3, last 2 paragraphs – Are they well located inside SF scaffolds section?
- Section 2.1 and 2.2 finishes with the authors' perception; it should be seen in the other sections.
- Page 7. NFkB acronym is not explained before.
- Page 8. It is biproducts or byproducts?
- The CONCLUSIONS answer the study's goals.
My recommendation would be to Accept, with the revisions noted above, to improve the manuscript.
Round 2
Reviewer 1 Report
The authors have replied to almost all the comments, and it has been improved in the revision. I recommend one minor correction before considering for publication in “MDPI-Pharmaceutics”
- Please include the suitable references in Table 1 and accordingly modify all the cited references in the manuscript.
Author Response
The authors would once again like to thank the reviewers for their time and thoughtful constructive criticisms. We have addressed the comment made in full. We detail the changes made in our response below. Please note that original reviewer comments are in standard black text, while the authors’ comments are italicized.
Reviewer 1:
The authors have replied to almost all the comments, and it has been improved in the revision. I recommend one minor correction before considering for publication in “MDPI-Pharmaceutics”
-
- Please include the suitable references in Table 1 and accordingly modify all the cited references in the manuscript.
- We agree that references should be included with the table, and have added them. Please see Table 1 on page 3 for a list of these references.